# Functionalized Metal Nanoparticles in Cancer Therapy

**DOI:** 10.3390/pharmaceutics15071932

**Published:** 2023-07-11

**Authors:** Paola Trinidad Villalobos Gutiérrez, José Luis Muñoz Carrillo, Cuauhtémoc Sandoval Salazar, Juan Manuel Viveros Paredes, Oscar Gutiérrez Coronado

**Affiliations:** 1Centro Universitario de los Lagos, Universidad de Guadalajara, Lagos de Moreno 47460, Mexico; paola.villalobos2452@academicos.udg.mx; 2Área de Ciencias de la Salud, Universidad Autonoma de Zacatecas, Zacatecas 98160, Mexico; mcbjlmc@gmail.com; 3División de Ciencias de la Salud e Ingenierías, Campus Celaya-Salvatierra, Universidad de Guanajuato, Celaya 38060, Mexico; cuauhtemoc.sandoval@ugto.mx; 4Centro Universitario de Ciencias Exactas e Ingenierías, Universidad de Guadalajara, Guadalajara 44430, Mexico; juan.viveros@academicos.udg.mx

**Keywords:** cancer, functionalized nanoparticles, metallic nanoparticles

## Abstract

Currently, there are many studies on the application of nanotechnology in therapy. Metallic nanoparticles are promising nanomaterials in cancer therapy; however, functionalization of these nanoparticles with biomolecules has become relevant as their effect on cancer cells is considerably increased by photothermal and photodynamic therapies, drug nanocarriers, and specificity by antibodies, resulting in new therapies that are more specific against different types of cancer. This review describes studies on the effect of functionalized palladium, gold, silver and platinum nanoparticles in the treatment of cancer, these nanoparticles themselves show an anticancer effect. This effect is further enhanced when the NPs are functionalized with either antibodies, DNA, RNA, peptides, proteins, or folic acid and other molecules. These NPs can penetrate the cell and accumulate in the tumor tissue, resulting in a cytotoxic effect through the generation of ROS, the induction of apoptosis, cell cycle arrest, DNA fragmentation, and a photothermal effect. NP-based therapy is a new strategy that can be used synergistically with chemotherapy and radiotherapy to achieve more effective therapies and reduce side effects.

## 1. Introduction

Cancer is one of the leading causes of death worldwide, both in developed and developing countries. Moreover, the number of deaths is expected to increase significantly, mainly due to three factors: the increase in population, age, and unhealthy lifestyles, which are considered risk factors for the development of this disease [1]. This disease includes a heterogeneous group of diseases defined by an abnormal and uncontrolled growth of the cells that make up the tissue, which can spread to other organs or tissues of the body [2,3].

This disease is the second leading cause of death worldwide [4], with estimates of 19.3 million new cases diagnosed and 10 million deaths worldwide [5]. Currently, treatment includes chemotherapy, radiotherapy, and surgery, which are considered conventional treatments [4]. However, in recent years, new treatments have been introduced, including stem cells, ablation therapy, targeted therapy, targeted chemotherapy, radiofrequency ablation, metallic nanoparticles or drug nanocarriers, and natural antioxidants (polyphenols and bioactive plant compounds), which are radical scavengers that treat or prevent cancer [6]. These new cancer therapies are being explored because conventional treatments, such as chemotherapy, cannot distinguish between tumor cells and normal cells. For example, radiation therapy mainly damages healthy tissue around the tumor and fails to completely kill cancer cells in large tumors. Similarly, surgical removal of the tumor does not always completely eliminate tumor tissue, leading to tumor recurrence and, in all cases, secondary effects of these treatments [7]. Therefore, scientific research is working to develop new methods and tools, such as nanotechnology, that can make cancer treatments more effective, less invasive, and more likely to eradicate the disease.

Currently, there is a variety of nanostructured systems that can be used for the development of nanomedicine in cancer therapy. These systems include both organic nanocarriers (polymers and liposomes) and inorganic nanocarriers (gold and silver) [8]. The different platforms offered by nanotechnology often reduce toxicity in normal tissues, increasing bioavailability, longer half-life, drug release, and biocompatibility. They exhibit properties that condition the improvement of therapeutic effects against cancer [9,10].

Nanoparticles (NPs) are particles less than 100 nm in size [11] and can be classified into three types depending on their composition: organic, carbon-based, and inorganic [12]. Organic NPs, such as dendrimers, micelles, liposomes, and ferritin, are biodegradable nanomaterials that are non-toxic and some have a hollow core [13]. Carbon-based NPs are composed of carbon and include fullerenes, graphene, carbon nanotubes, carbon nanofibers, and carbon-based quantum dots [14]. Finally, inorganic NPs are mainly composed of metals, metal oxides, and semiconductor materials [15].

Metallic NPs can be composed of different elements, such as platinum, palladium, gold, iron, silver, nickel, or copper (Pt, Pd, Au, Fe, Ag, Ni, Cu, respectively), which have various biomedical applications, e.g., in cancer treatment, immunotherapy, Magnetic Resonance Imaging (MRI)-guided gene therapy, MRI-guided thermal therapy, and MRI-guided chemotherapy. This effect can occur through various mechanisms such as local magnetic hyperthermia [16,17], specificity and optimized drug dosage [18], or improvement of radiotherapeutic performance [19]. Physicochemical properties such as shape, size, surface charge, and degree of dispersion can determine the biological effect of NPs. Noble metal NPs exhibit high corrosion and oxidation resistance and have lower cytotoxicity compared to transition metal NPs. Their high biocompatibility, ease of synthesis, surface functionality, and excellent optical properties make noble metal NPs (Pd, Ag, Au, and Pt) promising nanomaterials for targeted cancer therapy (Table 1).

## 2. Metallic Nanoparticles

Metallic NPs have a typical core/shell structure. The core is composed of a metal that determines the properties of the NPs, such as fluorescence, optical, magnetic, and electronic properties. The shell is composed of metals or organic polymers that protect the metallic core from chemical interactions with the environment and give the NPs the ability to be conjugated with some biomolecules [20]. These include: the low molecular weight ligands, peptides, proteins, polysaccharides, polyunsaturated and saturated fatty acids, deoxyribonucleic acid (DNA), plasmids, small interfering ribonucleic acid (siRNA), antibodies, tumor markers, and small molecules (Figure 1) [21]. Metallic NPs are often used as drug delivery systems or molecules that function as therapeutic agents through functionalization [22]. Recently, nanoparticle-based recombinant RNA and DNA delivery has been used in gene therapy, mainly because of its high transfection efficiency, low immunogenicity, biocompatibility, and most importantly, protection of genetic material from enzymatic degradation [23]. The entry of NPs into the cellular cytoplasm is very important for them to exert their effects. These nanostructures can be internalized by clathrin-mediated endocytosis, lipid raft/caveolae-mediated endocytosis, and micropinocytosis. The interaction that must take place between the NPs and the cell is very important. Some factors depend on this interaction, such as size, shape, surface charge, lipophilicity, type of nanocarrier, and finally the cell involved in the internalization [24]. The size of the NPs is an important factor in the internalization process because NPs with diameters between 10 and 100 nm are internalized via clathrin and via caveolae [25] to be incorporated into the intracellular traffic.

### 2.1. Palladium Nanoparticles

Pd is a noble metal and has particular characteristics, such as high thermal and chemical stability, catalytic activity, and an adjustable optical response, so it has high potential for radioactive image acquisition and photothermal therapy, suggesting a limited application in biomedicine [26]. However, some studies suggest that palladium nanoparticles (PdNPs) have therapeutic potential in cancer through photothermal and photodynamic therapies because this material absorbs in Near-Infrared spectroscopy (NIR), so strategies have been developed using NIR laser photoabsorbers to generate heat under NIR laser irradiation, resulting in highly specific cancer therapy. For example, Thapa et al. (2018) used PdNP-decorated graphene oxide (GO) (GO-PdNPs) to treat a solid prostate cancer tumor in a murine model, observing that intratumorally administered GO-PdNPs exerted a high photothermal effect, i.e., generation of reactive oxygen species (ROS). This is due to the high retention of GO-PdNPs in the solid tumor and the property of GO to absorb in the near-infrared region, which induces photoelectron interactions to generate heat, leading to photothermal ablation of the tumor [27]. Another study showed that these nanoparticles functionalized with trimethyl chitosan (TMC) enhanced the cytotoxic effect as well as the biocompatibility of PdNPs on breast cancer cells (MDA-MB-231 cells) and served as a photothermal agent in cancer therapy [28]. In another study, the effect of PdNPs coated with chitosan oligosaccharide (COS) and functionalized with arginine–glycine–aspartic acid (RGB) peptide was investigated on MB-231 breast cancer cells. It was shown that PdNPs functionalized with RGB peptide exerted enhanced photothermal therapeutic effects upon irradiation with an 808 nm laser and were undetectable in the tumor at day 20. It has also been shown that the specificity of PdNPs functionalized with the RGB peptide increased, as they accumulated more in the tumor, allowing greater specificity when phototherapy is applied with an NIR laser [29].

Recently, the combination of photothermal therapy and chemotherapy has been investigated to have a synergistic therapeutic effect, as photothermal therapy enhances drug uptake at the cellular level, which may help inhibit mechanisms of drug resistance. In this sense, a multifunctional system combining chemotherapy and photothermal therapy was developed. By adsorbing doxorubicin (DOX) on the surface of Pb nanosheets and functionalizing it with reduced glutathione (GSH), this conjugation significantly improved the accumulation of Pd nanosheets in tumor tissue as well as its absorption. On the other hand, the Pb nanosheets can be used as pH-dependent drug carriers to make the release of DOX more efficient. This nanostructured system enabled effective cancer therapy with a lower drug concentration and lower NIR laser power to achieve tumor shrinkage [30]. In another study, a platform of pH-sensitive PdNPs functionalized with transferrin and paclitaxel (PbNPs-PTX-Tf) was developed. This system combines photothermal therapy exerted by PdNPs, enhanced capture efficiency by transferrin, and a chemotherapeutic agent such as paclitaxel. When PbNPs-PTX-Tf were examined on MCF-7 and MDA-MB-231 cells, they showed a stronger inhibitory effect on cell viability than the system functionalized with transferrin. Moreover, treatment with PbNPs-PTX-Tf followed by NIR laser exposure resulted in a higher cytotoxic effect on both cancer cell lines. On the other hand, in vivo evaluation showed inhibition of tumor development due to accumulation of PbNPs-PTX-Tf at the tumor site, and this effect was enhanced with exposure to the NIR laser, suggesting that there is a cytotoxic synergist of paclitaxel and photothermal therapy [31]. In addition, Zhang et al. (2019) synthesized a system of hydroxy boron nitride nanosheets sensitive to stimuli as transporters for PdNPs and drugs such as doxorubicin (Pd-OH-BNNS/DOX). This system was evaluated in MCF-7 cells and showed inhibition of cell proliferation, whereas the nanosheets without doxorubicin (Pd-OH-BNNS) showed no cytotoxic effect. The combined evaluation of Pd-OH-BNNS/DOX and photothermal therapy in an in vivo model showed higher antitumor efficacy than that exerted with Pd-OH-BNNS alone. This effect suggests that PdNPs absorb light and convert it into heat, which induces DOX release to enhance the anticancer effect [32].

Functionalization of Pd-based nanomaterials has been shown to be very useful as it can increase the specificity of PdNPs and increase the concentration of these nanomaterials in the tumor, allowing for a more specific therapy with very high efficacy in the treatment of cancer.

### 2.2. Gold Nanoparticles

Gold has a high atomic number, which causes gold nanoparticles (AuNPs) to absorb more photons than soft tissue [33], have a wide range of applications in both the diagnosis and treatment of cancer due to their high bioavailability and biocompatibility, ease of synthesis and bioconjugation, and chemical and physical properties [34,35,36]. Particularly important features include localized surface plasmon resonance (LSPR), radioactivity, and high X-ray absorption coefficient. However, surface-enhanced fluorescence (SEF), photothermal transformation, photochemical transformation, and colorimetric reactions stand out in diagnosis [37]. In addition, its physical properties such as size and shape can affect absorption at the cellular level, leading to better results when incorporated into nanospheres, which have this ideal shape due to their high surface-to-volume ratios and low toxicity [33].

In addition, their chemical properties have the ability to form stable chemical bonds with groups containing sulfur and nitrogen (S and N, respectively) in their structure. Through these bonds, the surface of AuNPs can be functionalized with a variety of ligands, such as antibodies [38,39,40], proteins [41,42,43], peptides [44,45], and other biomolecules to obtain biologically active conjugates.

Currently, drugs that induce apoptosis by generating ROS are important in cancer therapy. In this sense, Khiavi et al. (2020) synthesized AuNPs and functionalized them with RNase A and stabilized them with polyethylene glycol (PEG) (AuNPs-RNase-PEG) and then investigated their effect on colon cancer cells (SW-480). Treatment with AuNPs-RNase-PEG showed a cytotoxic effect mainly associated with chromatin condensation and nuclear fragmentation, eventually leading to apoptosis. In addition, increased levels of ROS were detected, suggesting that there is a correlation between the production of ROS and the induction of apoptosis [46]. Antibody-based therapy for malignant tumors has recently increased. This is mainly due to their high specificity. With the addition of antibodies, AuNPs become more selective and target more specifically. For example, functionalization of these NPs with a specific monoclonal antibody for HER2-positive breast cancer (trastuzumab) has been shown to be an effective strategy in the treatment of cancer. AuNPs functionalized with this antibody and stabilized with polyamidoamine (AG-MA1-SH) have shown high efficacy in HER2 cells overexpressing SKBR-3 cells. This effect is mainly mediated by an increase in pro-apoptotic proteins, a decrease in anti-apoptotic components, and of survival and proliferation pathways down regulation [47]. The use of these therapies based on AuNPs functionalized with trastuzumab is applied to some cancers that develop resistance to trastuzumab treatment, including gastric cancer. In vitro studies have shown that AuNPs functionalized with trastuzumab have a cytotoxic effect on gastric cancer cells that are sensitive or resistant to trastuzumab (NCI-N87 and MKN7, respectively). This effect is mainly mediated by the induction of autophagy and oxidative stress [48]. Another important therapeutic target is the epidermal growth factor receptor (EGFR), a transmembrane protein responsible for triggering signaling cascades associated with proliferation, angiogenesis, apoptosis, and metastasis; its overexpression is associated with the progression of some cancers. AuNPs functionalized with anti-EGFR (cetuximab) (C-AuNPs) have a stronger cytotoxic effect than cetuximab alone. This effect was evaluated in HT-29 colorectal cancer cells, this functionalization increases the endocytic capacity and thus the degradation of the EGFR by the lysosome, and this degradation becomes relevant in cancer therapy as it inhibits cell proliferation and induces apoptosis in HT-29 cancer cells [49]. In another study, the cytotoxic effect of C-AuNPs as carriers of 5-fluorouracil on colorectal cancer cells (HCT-116 and HT-29) was investigated. The combination of the antibody and anticancer drug-induced apoptosis and necrosis as well as inhibition of cell proliferation in the cancer cell lines compared with C-AuNPs suggest that the chemotherapeutic effect of 5-fluorouracil is enhanced by NPs [50]. Another study demonstrates the in vitro and in vivo antitumor effects of C-AuNPs on the EGFR highly expressing non-small cell lung cancer (NSCLC). These NPs significantly inhibit cell proliferation, decrease cell migration, and increase induced apoptosis by cetuximab. In the in vivo model, these NPs cause a reduction in tumor growth. These effects are more pronounced in EGFR^high^ A549 cells than in EGFR^low^ H1299 cells, and the mechanism of this effect is mainly due to the enhanced internalization of cetuximab promoted by AuNPs [51]. The antitumor effect is mainly associated with the ERK/MAPK and PI3K/Akt signaling pathways. These are two important signaling pathways downstream of EGFR and have been associated with proliferation, apoptosis, and differentiation processes in NSCLC [52].

Using folate-conjugated AuNPs, Jin et al. (2012) investigated the photothermal effect of these nanostructures in human hepatocarcinoma HepG2 cells. The effect of the folate-conjugated nanostructures is to inhibit the growth of cancer cells. This inhibition is mainly associated with apoptosis and structural changes in the membrane and its composition and function, which alters intracellular Ca^2+^ homeostasis by triggering photothermal therapy-mediated apoptotic signals in HepG2 cells [53]. In another study, the effect of intense pulsed light (IPL) and two conjugated folate AuNPs on HeLa and MCF-7 cells was investigated. This showed that both the functionalized NPs and IPL did not alter cell viability. However, when the treatment (AuNPs and IPL) is combined, cell death is up to 98% in HeLa cells and 9% in MCF-7 cells, demonstrating the specificity of the cytotoxic effect only on cells overexpressing the folate receptor through the receptor-mediated enhanced uptake of folate-conjugated NPs [54]. Molecules for cancer treatment include siRNAs, also known as short interfering RNA or silencing RNA molecules, which can selectively down regulate genes involved in disease pathology. These siRNAs have a short half-life mainly due to enzymatic degradation, so conjugation with NPs may be a strategy to develop treatment through these siRNAs. In this sense, Minassian et al. (2023) used AuNPs functionalized with folic acid and siRNA (AuNPs-FA-siRNA) as carriers to deliver siRNAs into prostate cancer cells via folate receptors. By functionalizing AuNPs, it was possible to observe siRNAs at the intracellular level, whereas no siRNA was observed at the intracellular level for AuNPs that were not conjugated to folate receptors [55]. Carriers are important for the release of siRNAs in a solid tumor; in this sense, Kim et al. (2014) developed siRNA-functionalized AuNPs (siRNA-AuNPs) and evaluated luciferase gene silencing in HeLa cells expressing luciferase (HeLa-Luc) in an in vivo model. This study showed that siRNA-AuNPs increased circulation time at the systemic level compared with unconjugated siRNAs; furthermore, there was a greater accumulation of functionalized NPs in the cervical cancer model when HeLa-Luc cells were subcutaneously inoculated. This silenced the luciferase gene in tumor tissue and reduced the intensity of luminescence by up to 50%, suggesting that siRNA-AuNPs efficiently release siRNAs in tumor tissue [56]. Through this strategy, siRNA-based therapy can be implemented to be specifically delivered to the tumor and incorporated into the cytoplasm to modulate gene expression or silence genes in cancer cells. For example, vascular endothelial growth factor (VEGF) siRNAs and B-cell lymphoma/leukemia-2 (Bcl-2) siRNAs, which were used to functionalize gold nanoparticles entrapped in dendrimers for gene silencing applications on U87-MG cells in vitro and the xenografted tumor model in vivo [57].

The development of nanocarriers to transport anticancer molecules is a promising system for the treatment of cancer; in this sense, Giraldez et al. (2022) developed a transport system for doxorubicin (Doxo) based on a biocompatible gold nanosystem. This system consists of AuNPs covered with a cationic gemini surfactant (16-Ph-16), which plays an important role in minimizing electrostatic repulsion and in compacting the DNA-Doxo complex to form a compacted nanocomplex (Au@16-Ph-16/DNA/Doxo). The purpose of using DNA as a biopolymer is to maintain the integrity of the nanostructure and improve biocompatibility. When this nanosystem was evaluated on prostate cancer cells (LNCaP; DU-145) and liver cancer (SNU387), it showed a significant cytotoxic effect, even exceeding that of the drug Doxo. Also, when a formulation of Au@16-Ph-16/DNA/Doxo + Au@16-Ph-16 was evaluated, it showed a more significant effect on DU-145 cells and a small effect on normal liver and prostate cells (HepG2 and PNT2, respectively), suggesting that this nanosystem utilizes cancer cell properties to exert a more effective cytotoxic effect than the Doxo treatment [58]. Another system uses two biocompatible gold nanosystems: Au@16-Ph-16 and a DNA/5-fluorouracil complex (Au@16-Ph-16/DNA-5-Fu). This nanosystem was studied in lung cancer cells (A549) and healthy cells (BJ-hTERT). This showed a cytotoxic effect on A549 cells when using the nanosystem with a higher concentration of reactants (5-Fu = 7.5 μM, DNA = 8.9 μM, and Au@16-Ph-16 = 0.33 nM) and a mild cytotoxic effect on BJ-hTERT cells, indicating that this nanocarrier can transport concentrations of 5-Fu and penetrate into the cell to release the drug into the cells at the target site and exert a cytotoxic effect on cancer cells [59]. The use of these nanocomplexes leads to reversible compaction of the DNA–drug complex; so, these nanosystems with DNA as a biopolymer and the cationic surfactant 16-Ph-16 represent a strategy to induce the release of anticancer drugs from compacted nanosystems.

On the other hand, alteration of the tumor microenvironment by cytokines is a promising therapy against cancer; however, a limitation in the implementation of this therapy is the counter-regulatory mechanisms. Therefore, Corti et al. (2021) developed gold nanospheres coated with the peptide isoDGR (isoAsp-Gly-Arg), functionalized with tumor necrosis factor-α (TNF) and/or interleukin-12 (IL12), and investigated their anticancer effects in a murine model of fibrosarcoma. The nanoparticles with the peptide and functionalized with TNF (iso1Au/TNF) and isoAu/TNF+IL12 show an antitumor effect in the in vivo model of fibrosarcoma by reducing the tumor volume. They also studied the synergistic effect of TNF (iso1Au/TNF) and isoAu/TNF+IL12 with Doxo, and they found that the antitumor effect was enhanced, which was due to the increased endothelial permeability and penetration of the drug into the tumor, improving the antitumor activity of Doxo [60].

AuNPs have properties such as photothermal conversion efficiency, long-lasting photostability, colloidal stability, biocompatibility, and uptake in cancer cells. In addition, their chemical and biological stability, size, and biocompatibility make AuNPs candidates for the evaluation of their photothermal effect.

The ability of NPs to convert light to heat has been studied to develop noninvasive therapies. In this sense, Guglielmelli et al. (2021) developed keratin-functionalized AuNPs (Ker-AuNPs), a biocompatible coating. In this study, the biocompatibility of these NPs and their PPT effect on the human glioblastoma cell line (U87-MG) were investigated. It showed 100% biocompatibility even at a concentration of 50 µM for 72 h. Subsequently, cells were exposed to laser illumination (10 min at 11.3 W/cm^2^) in the presence of Ker-AuNPs, showing a cytotoxic effect due to the temperature rise compared to cells without NPs [61]. In another study, the photothermal effect of spherical and anisotropic AuNPs of different sizes and FA-conjugated (FA-AuNPs) on HeLa cells was investigated. This shows that the NPs do not alter cell viability and even promote proliferation. This effect has been associated with FA as it promotes cell proliferation. However, when FA-AuNPs are irradiated with a laser of 638 nm at 600 mW for 5 min, the effect that spherical FA-AuNPs exert on cell viability at a concentration of 50 uM is 79% and 75% for the NPs with sizes of 9 and 14 nm, respectively. When the anisotropic FA-AuNPs with sizes of 15 and 20 nm were evaluated, cell viability decreased to 66% and 61%, respectively. This study shows that FA-AuNPs induce cell death by apoptosis, and the cytotoxic effect of larger NPs is evident, suggesting that the photothermal effect exerted by NPs can be modulated according to size and shape to achieve a more efficient cytotoxic effect [62].

AuNPs are ideal to be functionalized with antibodies. They can also act as drug carriers, and their photothermal effect and biocompatibility make these AuNPs a promising nanomaterial for the development of new therapeutic strategies against cancer.

### 2.3. Silver Nanoparticles

The biological activity of silver nanoparticles (AgNPs) depends on factors including: surface chemistry, size, shape, morphology, composition, and reactivity in solution. These properties are crucial factors for their cytotoxic activity [63]. Another important feature is their ability to conjugate different types of ligands to their surface to increase their specificity and exert an effect [64]. Currently, the functionalization of AgNPs is aimed at developing targeted treatments to improve their effect on target cells. Functionalization includes antibodies [65,66], carbohydrates [67,68], polysaccharides [69], proteins [70,71], folic acid [72,73,74], and other molecules. AgNPs have shown high potential as anticancer agents due to their nanometric nature. These nanoparticles can actively or passively penetrate tumor tissue, where they can accumulate. Passive accumulation is due to the particular cytoarchitecture of tumor tissue, which includes an atypical endothelial layer and the absence of tissue-forming vessels and pores between 100 nm and 2 µm in diameter, resulting in the accumulation of nanometer-sized materials in cancer tissue. This phenomenon is referred to as the enhanced permeability and retention (EPR) effect [75]. On the other hand, active targeting has been proposed to increase the specificity of AgNPs against cancer. This specificity is based on a biological interaction between the ligands on the surface of the NPs and the target cell [76].

The cytotoxic effect of AgNPs is mainly due to the generation of ROS and oxidative stress, DNA damage, cell cycle arrest, and induction of tumor cell death by apoptosis [77,78,79]. The uptake of AgNPs by cells increases the synthesis and accumulation of ROS at the intracellular level, resulting in damage to various cellular components, degradation of DNA, lipid peroxidation, and carbonylation of proteins, causing oxidative stress-induced apoptosis and damaging the components of the cell [80,81]. Apoptosis is triggered by the release of Ag ions in the cytosol, which is promoted by the acidic lysosomal environment and increases the permeability of the mitochondrial membrane, leading to mitochondrial dysfunction and eventually cell apoptosis [82]. Several studies suggest that AgNPs are able to affect the expression of proteins that regulate the cell cycle. These proteins include cyclins B, E, and D, as well as up-regulation of p53 phosphorylation, leading to cell cycle arrest in the G2/M phase and increased production of ROS and the eventual induction of apoptosis [83].

The generation of anticancer therapeutic targets by functionalizing AgNPs has led to the development of specific agents with antitumor potential and lower toxicity to normal tissues. In one study, functionalization of AgNPs with monoclonal antibodies targeting EGFR (AgNPs-Anti-EGFR) was investigated in human nasopharyngeal carcinoma epithelial (CNE) cells and found that AgNPs-Anti-EGFR inhibited cell growth, induced apoptosis, increased sensitivity to radiation, and decreased expression of DNA damage/repair proteins such as Ku-70, Ku-80, and Rad51. This suggests that AgNPs-Anti-EGFR have a radiosensitizing effect as well as an antiproliferative effect on CNE cells [84]. Another study demonstrated the efficacy of affibody Z_HER2:342_-conjugated AgNPs (AgNPs-HER2) in cancer therapy by investigating these nanoparticles on HER2 overexpressing cancer cells and on a xenograft tumor model. Human ovarian adenocarcinoma SKOV3-1ip cells overexpress HER2 and Chinese hamster ovary (CHO) cells as negative control. These results showed that the binding of AgNPs-HER2 in SKOV3-1ip cells compared with CHO cells was 10.6-fold higher. The interaction between cancer cells with AgNPs-HER2 and irradiation LED matrix (465 nm; power of 95 mW/cm^2^) exerts a cytotoxic effect, which is mainly caused by the increased production of ROS. However, cytotoxicity assays do not reflect the effect in an in vivo model. When AgNPs-HER2 followed by light irradiation was evaluated on the solid tumor, 100% inhibition of tumor growth was demonstrated, and metastasis was also inhibited [85].

On the other hand, conjugation of AgNPs with bovine serum albumin (BSA) (AgNPs-BSA) exerts a cytotoxic effect on B16F10 melanoma cells, which is mainly caused by oxidative stress. In addition to this effect, the nanoparticles inhibit angiogenesis. This was evaluated by the tube formation assay and a typical wound-healing assay, where it was observed that these BSA-conjugated materials inhibited VEGF-induced tube formation and cell migration, suggesting that AgNPs-BSA are able to inhibit angiogenesis [86]. In another study, the effect of albumin-conjugated nanoparticles (AgNPs-A) was investigated in a breast cancer cell line (MDA-MB-231 and MCF-7). This showed a cytotoxic effect of AgNPs-A by fragmenting DNA, ultimately leading to apoptosis of the cells; however, the nanoparticles did not show any effect on normal cells (MCF-10A). The production of ROS leads to a change in the homeostasis of the antioxidant system of cells. AgNPs-A produce high levels of ROS, suggesting that cell death is mediated by intrinsic apoptotic pathways, with this effect induced by the increased production of ROS [87]. Under stress conditions, as in tumor tissue, the cancer cell takes up albumin as a source of energy and amino acids to a greater extent than normal cells. This increased uptake possibly occurs via specific albumin (albondin) receptors [88], so that albumin-functionalized nanoparticles are taken up by tumor cells to a greater extent than by normal cells, thus exerting their cytotoxic effects on cancer cells.

The conjugation of molecules such as tumor necrosis factor-related apoptosis-inducing ligand (TRAIL) on nanoparticles may be of interest in the treatment of tumor cells resistant to chemotherapy. TRAIL belongs to a group of the tumor necrosis factor (TNF) family that is involved in the apoptosis of cancer cells via the death receptors DR4 and DR5 without causing cytotoxicity in normal cells, making it a potential antitumor agent [89]. Despite its advantages, its applicability is limited, mainly due to the resistance that cancer cells develop to TRAIL-mediated apoptosis. Thus, in a study conducted by Sur-Erdem et al. (2020), the effect of TRAIL conjugated with AgNPs (TRAIL-AgNPs) on T98 G TRAIL-sensitive (TS) and T98 G TRAIL-resistant (TR) cells was investigated. Cell viability of TS cells exposed to TRAIL was 8.4% at a dose of 500 ng/mL. However, when cells were exposed to TRAIL-AgNPs, cell viability decreased to 5.9%, demonstrating that functionalization of NPs with TRAIL increases cytotoxic effects. In TR cells, exposure to TRAIL-AgNPs decreased cell viability by up to 44.3% at the same dose of 500 ng/mL, whereas treatment of these cells with only TRAIL and AgNPs did not affect cell viability. In addition, caspases eight and nine activity was increased in both TS and TR cells treated with TRAIL-AgNPs. This study suggests that functionalization of TRAIL with AgNPs increases sensitization of cells resistant to TRAIL [90].

Another potential target in the treatment of resistant forms of cancer is glucose uptake because cancer cells preferentially metabolize glucose by glycolysis and lactate fermentation to gain energy rapidly (Warburg effect), but this is energy-inefficient. Therefore, cancer cells require a high rate of glucose uptake to meet the increased energy demand during tumor progression [91]. In this sense, Morais et al. (2022) functionalized AgNPs with glucose (G-AgNPs) and investigated their effect on prostate cancer cells (PCa) resistant to androgen deprivation therapy (ADT) (DU-145 and PC-3). The effect of G-AgNPs on the proliferation of DU-145 and PC-3 cells achieved an inhibition of cell proliferation of 14.12% and 19.21%, respectively, while AgNPs alone inhibited the proliferation of DU-145 to 27.21% and of PC-3 to 25.68%, indicating that AgNPs functionalized with glucose have a stronger effect on the proliferation of cancer cells. This mainly arrest-mediated effect is S-phase, indicating DNA damage associated with the formation of ROS and mitochondrial damage, which was ultimately observed by inhibiting cancer cell proliferation and apoptosis [92]. This cytotoxic effect of G-AgNPs was also observed in HepG2 and neuron 2A cells. This effect has been linked to oxidative stress associated with the formation of ROS assessed by protein carbonylation [93].

### 2.4. Platinum Nanoparticles

Pt is a noble metal with significant catalytic activity and has anticancer activity itself [94]. Platinum nanoparticles (PtNPs) have recently gained attention due to their wide range of applications such as biosensor, electro-analytical and analytical, and catalysis, standing out in biological applications [95]. These materials have catalytic [96] and enzymatic activity and are inherently biocompatible, suggesting that they can be used as anticancer agents [97]. PtNPs have been shown to possess some intrinsic cytotoxic activity that slows tumor growth. However, the specificity is enhanced by functionalization of the nanoparticles, which increases their anticancer cytotoxic capacity [98].

Recently, the cytotoxic effect of a system of octopod-shaped PtNPs functionalized with doxorubicin (PtNPs-DOX) was demonstrated by the release of the chemotherapeutic agent in cancer cells (MCF-7 and MDA-MB-231). The effect is mediated by activation of the tumor suppressor gene (PTEN), which restricts the PI3K/AKT signaling pathway, leading to mitochondrial dysfunction and activation of caspases three and nine, ultimately resulting in cell apoptosis [99]. In another study, a PEGylated PtNPs nanoconjugate was synthesized as a DOX transport system (PtNPs-DOX), and its effect on human lung cancer and mouse melanoma cells (B16F10 and A549, respectively) was investigated. These nanoconjugates showed stronger inhibition of proliferation in both cell lines. They also showed sub-G1 phase and G2/M cell cycle arrest, as well as a higher apoptosis rate in cancer cells compared to DOX alone. On the other hand, in vivo evaluation in a subcutaneous melanoma model in mice showed a reduction in tumor volume and weight in animals treated with PtNPs-DOX. These results show positive regulation of p53 and caspase three, confirming the induction of apoptosis, while STAT3 EGFR and Bcl-2 are down regulated, suggesting that p53 is involved in the anticancer effect of PtNPs-DOX [100].

Functionalization of PtNPs with biopolymers is a therapy directed against tumors. It is known that some tumor cells overexpress receptors for hyaluronic acid (HA) (CD44) [101]. On this basis, an in vitro study showed that PtNPs encapsulated in hyaluronic acid (HA-PtNPs) accumulate in tumor cells, overexpressing CD44 (MDA-MB-231 cells) when compared with nonfunctionalized PtNPs and in cells not expressing CD44 (NIH3T3 cells). However, this accumulation has no cytotoxic effects. On the other hand, the effect of HA-PtNPs in combination with directed photothermal therapy (PTT) was investigated in an in vitro assay. This showed a cytotoxic effect on MDA-MB-231 cells, suggesting that PTT in combination with HA-PtNPs is more efficient in cells overexpressing CD44. Moreover, an in vivo study in mice with MDA-MB-231 breast tumors showed that intravenous administration of HA-PtNPs can be detected in the tumor and that after NIR irradiation, the temperature in the tumor is increased, leading to a decrease in tumor size [102]. In addition, the functionalization of PtNPs with polyethylene glycol (PEG) (PEG-PtNPs) in HeLa cells was investigated. The NPs show distribution in the cytoplasm of cancer cells, and associated with this, an increased cytotoxic effect was found upon irradiation with γ-rays compared to cells without NPs, suggesting that PEG-PtNPs enhance the effect of radiation on cancer cells and thus exert a greater cytotoxic effect [103].

Another study proposes the functionalization of PtNPs with BSA (BSA-PtNPs) to improve radiotherapy because BSA acts as a carrier, in this case of PtNPs. In this sense, BSA-PtNPs were studied in mouse breast cancer cells (4T1) and irradiated with X-rays. The results showed a greater cytotoxic effect compared to cells treated with only the BSA-PtNPs, suggesting that these nanomaterials may act as a potential radiosensitizer by improving the efficacy of radiotherapy [104]. In addition, Yaray et al. (2023) conjugated methotrexate (MTX) to BSA-PtNPs (BSA-PtNPs-MTX) to perform chemoradiotherapy. The effect was evaluated in 4T1 cells and showed that BSA-PtNPs-MTX had a cytotoxic effect. When these cells were additionally irradiated with X-rays, the effect was even stronger and was mainly mediated by the generation of apoptosis. These results suggest that the effect of functionalized PtNPs increases ROS production after radiotherapy and enhances the apoptosis index, which provides greater efficacy in cancer treatment [105].

Finally, the use of molecules such as folic acid on nanocarriers directs therapy to cancer cells that overexpress folate receptors. Therefore, the design and synthesis of folic-acid-coated PtNPs (FA-PtNPs) have provided important results in terms of cytotoxicity, showing increased absorption and a cytotoxic effect on HeLa, MCF-7, and IMR90 cells. This effect is mainly mediated by necrosis and apoptosis. These results demonstrate an inherent cytotoxicity of FA-PtNPs [106].

**Table 1 pharmaceutics-15-01932-t001:** The role of NPs in cancer treatment.

Nanoparticle Type	Nanoparticle Size (nm)	Funtionalization Material	Type of Cancer	Mechanism of Action	Zeta Potential (mV)	References
PdNPs-decorated Graphene oxide	174.8 ± 5.8	Graphene oxide	Solid prostate cancer tumor	Photothermal effect, ROS generation	−17.5 ± 0.6	[27]
Trimethyl-chitosan-coated PdNPs	69.9 ± 6	Trimethyl chitosan	Breast cancer cells	Photothermal effect	12.8 ± 6.1	[28]
COS-coated PdNPs functionalized with the RGB peptide	24.28 ± 1.29	RGB peptide	Breast cancer cells	Photothermal effect	14.65	[29]
Pb nanosheets functionalized with DOX and GSH	4.4	Doxorubicin and GHS	Hepatoma cells and breast cancer cells (in vitro and in vivo)	Chemotherapy and photothermal effect	7.9	[30]
PdNPs functionalized with PTX and Tf	164.6 ± 8.7	PTX and Tf	Breast cancer cells (in vitro and in vivo)	Chemotherapy and photothermal effect, DNA damage, G0/G1 phase cycle arrest, ROS generation	−13.2 ± 1.8	[31]
Nanosheet-decorated PbNPs and DOX	200	Doxorubicin	Breast cancer cells (in vitro and in vivo)	Chemotherapy and photothermal effect	−18.36 ± 0.52 **	[32]
AuNPs functionalized with RNase A-PEG	36	RNase A	Colon cancer cells	Chromatin condensation and nuclear fragmentation, apoptosis, ROS generation	−23.4	[46]
AuNPs functionalized with trastuzumab	5; 85.39 ± 0.68	Anti-HER2	Breast cancer cells and gastric cancer cells (in vitro and in vivo)	Apoptosis, survival–proliferation pathways down regulation, autophagy, oxidative stress	32.6 ± 9.4; −39.43 ± 0.85	[47,48]
AuNPs functionalized with cetuximab	60; 14	Anti-EGFR	Colorectal cancer cells, EGFR highly expressing non-small cell lung cancer (in vitro and in vivo)	Inhibit cell proliferation, apoptosis, inhibit cell migration and cell angiogenic activity, apoptosis	−27.7; −20.4	[49,51]
5-FU carried by AuNPs functionalized with anti-EGFR	18.83 ± 1.52	5-fluorouracil, anti-EGFR	Colorectal cancer cells	Apoptosis	−33.1 ± 3.78	[50]
Folate-conjugated Au nanorods	450–500	Folate	Hepatocellular carcinoma cells	Apoptosis, alter intracellular Ca^2+^ homeostasis, photothermal effect, nuclei fragmentation	---	[53]
Folate-conjugated AuNPs	4.8	Folate	Cervical cancer cells and breast cancer cells	Photothermal effect	---	[54]
Au nanosystems covered with DOX/DNA	67 ± 11; 71 ± 11; 64 ± 9 ***	Doxorubicin/DNA	Hepatocellular carcinoma cells, prostate cancer cells	Enhance the effect of the anticancer agent	−28 ± 5, −44 ± 5, −30 ± 5 ***	[58]
Au nanosystems covered with DNA/5-FU	34 ± 7; 85 ± 5; 14 ± 3 ***	DNA/5-fluorouracil	Lung cancer cells	Enhance the effect of the anticancer agent	−45 ± 8; −46 ± 6; −56 ± 5 ***	[59]
Trifunctional AuNPs	27.5 ± 2.3	isoDGR, IL12, TNF	Fibrosarcoma cells (in vivo)	Increase tumor vascular permeability, enhance the effect of the anticancer agent	---	[60]
Keratin-coated AuNPs	32.7	Keratin	Glioblastoma cells	Photothermal effect	−18 ± 2	[61]
Folic-acid-conjugated AuNPs	9, 14; 15, 20	Folic acid	Cervical cancer cells	Apoptosis, photothermal effect, ROS generation	−18.0, 28.9; −15.0, 21.2	[62]
Anti-EGFR-functionalized AgNPs	15 and 45	Anti-EGFR	Nasopharyngeal epithelial carcinoma cells	Radiosensitizing effect, antiproliferative effect, apoptosis, decreased expression of DNA damage/repair proteins	---	[84]
Anti-HER2-functionalized AgNPs	35.4 ± 1.6	Affibody Z_HER2:342_	Ovarian adenocarcinoma cells, mammary gland carcinoma cells (in vitro and in vivo)	Photothermal effect, ROS generation	---	[85]
BSA-coated AgNPs	100	BSA	Melanoma cells	Oxidative stress, inhibit angiogenesis, photothermal effect	−31.3 ± 5.3	[86]
Albumin-coated AgNPs	90	Albumin	Breast cancer cells	Fragmenting DNA, apoptosis, ROS generation	−20	[87]
TRAIL-conjugated AgNPs	106	TRAIL	Glioblastoma cells	Caspases 8 and 9 activity increased	---	[90]
Glucose-functionalized AgNPs	100	Glucose	Prostate cancer cells resistant to androgen deprivation therapy	ROS generation, mitochondrial damage, apoptosis, oxidative stress	−50 ± 4	[92]
DOX-functionalized octopod-shaped PtNPs	231 ± 1.1	Doxorubicin	Breast cancer cells	Activation of the tumor suppressor gene, mitochondrial dysfunction, activation of caspases 3 and 9 and apoptosis	−12.5 ± 5.05	[99]
DOX-functionalized PEGylated PtNPs	5–20	Doxorubicin	Lung cancer, melanoma cells	Sub-G1 phase and G2/M cell cycle arrest, apoptosis, up-regulation of p53, down regulation of SOX2 and Ki67	28.1 ± 1.7	[100]
PtNPs encapsulated in hyaluronic acid	38 ± 6	Hyaluronic acid	Breast cancer cells (in vitro and in vivo)	Photothermal effect, apoptosis	31 ± 1	[101]
PEGylated Pt nanoflowers	34.8 ± 5.3	Polyethylene glycol	Cervical cancer cells	Enhance radiation effects	20 ± 2	[103]
BSA-functionalized PtNPs	29.4	BSA	Breast cancer cells	Radiosensitizer	−29.3	[104]
MTX-conjugated BSA-PtNPs	28	Methotrexate	Breast cancer cells	Radiosensitizer, ROS generation, apoptosis	−25	[105]
Folic-acid-coated PtNPs	91.3	Folic acid	Cervical cancer, breast cancer cells	Apoptosis, necrosis	−40.5	[106]

** Pd-OH-BNNS at 58.3 °C; *** Zeta potential and diameter in PBS.

## 3. Future Perspectives

Based on the work cited in this review, cancer therapy based on NPs can be considered new. However, there is a large number of studies targeting different types of cancer, demonstrating the efficacy of NPs and NPs functionalized with different biomolecules in cancer therapy. The therapeutic potential of NPs ranges from photothermal therapy, drug nanocarriers, ROS generation, induction of apoptosis, mitochondrial dysfunction, cell cycle arrest, DNA fragmentation, inhibition of cell migration, lipid peroxidation, modulation of the immune system through cytokines, to others. All studies show the in vitro and in vivo antitumor effects with promising results, even in tumors resistant to chemotherapeutic agents. However, further studies are needed that focus on the design of functionalized NPs based on markers expressed by cancer cells, pharmacological studies that ensure adequate distribution and have high selectivity in the target cell, and toxicological studies that demonstrate the interaction of NPs with normal organs and/or cells. This would help to achieve better results in cancer therapy and also to evaluate whether these new therapies based on nanomaterials have minimal adverse effects.

## 4. Conclusions

NPs are proposed as new specific treatment alternatives for some cancers, as some studies have shown a cytotoxic effect on cancer cells and solid tumors, through cell cycle arrest, oxidative stress, ROS production, DNA damage, or death by apoptosis. Functionalization of NPs with other biomolecules may be a useful strategy for targeted, specific, effective, and, most importantly, noninvasive cancer therapy in healthy tissues. In this review, we provide an overview of how functionalized NPs can help cancer treatment to be more effective and less toxic. Although some studies are still in the preclinical phase, progress shows that they have beneficial effects and will therefore be a useful tool for the treatment of cancer.

## Figures and Tables

**Figure 1 pharmaceutics-15-01932-f001:**
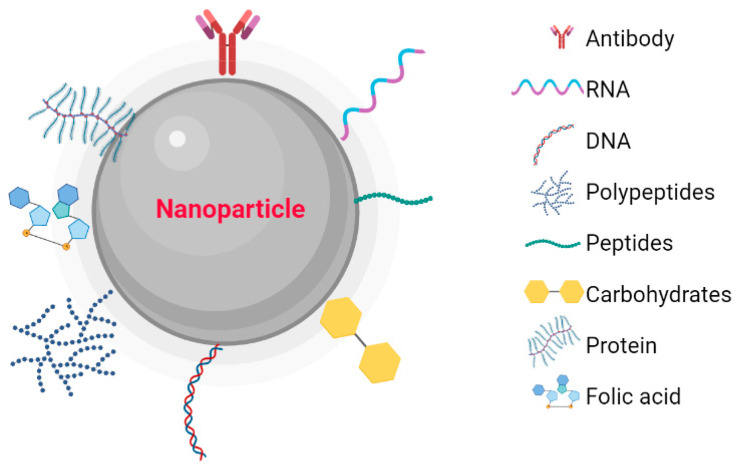
Nanoparticles functionalized with biomolecules: Antibodies, RNA, DNA, Polymers, Peptides, Carbohydrates, Proteins, and Folic Acid. Created with BioRender.com.

## Data Availability

Not applicable.

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
