# Peer review of "Functionalized Metal Nanoparticles in Cancer Therapy"

_pharmaceutics, 2023, doi:10.3390/pharmaceutics15071932_

Round 1

Reviewer 1 Report

The manuscript entitled: “Functionalized metal nanoparticles in cancer therapy” by Paola Gutierrez and coworkers has been revised. The manuscript is well-written and sounds interesting for the scientific community of Pharmaceutics. However, some aspects of the manuscript should be improved and I feel that the paper needs a major revision before it can be published in the journal of Pharmaceutics.

 -In the introduction section the reason for selecting palladium, gold and silver nanoparticles among others metallic nanoparticles is needed to be included.

-Page 2, line 48-49. The definition of nanoparticles is incorrect. In fact, NPs can have a size that ranges from 1 nm to 100 nm. See for instance:

Arabian Journal of Chemistry (2019) 12, 908-931.

-In section 2, metallic nanoparticles, the authors have forgotten to add comment or mention to the important role of the size of nanoparticles in the cell internalization, toxicity and the protection against enzymatic degradation exerted to biomolecules like DNA or RNA. See for instance:

-Thorek, D.L.; Tsourkas, A. Size, charge and concentration dependent uptake of iron oxide particles by non-phagocytic cells. Biomaterials 200829, 3583–3590

-Ghosh, P.S.; Kim, C.K.; Han, G.; Forbes, N.S.; Rotello, V.M. Efficient gene delivery vectors by tuning the surface charge density of amino acid-functionalized gold nanoparticles. ACS Nano 20082, 2213–2218

-In section 2.2 about the role of functionalized gold nanoparticles in cancer therapy, the authors have forgotten some recent important works. A comment about minimization of side effects of functionalized gold nanoparticles and different nanoformulation is needed. See for instance:

- Giráldez-Pérez, R.M.; Grueso, E.; Montero-Hidalgo, A.J.; Luque, R.M.; Carnerero, J.M.; Kuliszewska, E.; Prado-Gotor, R. Gold Nanosystems Covered with Doxorubicin/DNA Complexes: A Therapeutic Target for Prostate and Liver Cancer. Int. J. Mol. Sci. 202223, 15575.

 -Giráldez-Pérez, R.M.; Grueso, E.; Domínguez, I.; Pastor, N.; Kuliszewska, E.; Prado-Gotor, R.; Requena-Domenech, F. Biocompatible DNA/5-Fluorouracil-Gemini Surfactant-Functionalized Gold Nanoparticles as Promising Vectors in Lung Cancer Therapy. Pharmaceutics 202113, 423.

Ahmad, T.; Sarwar, R.; Iqbal, A.; Bashir, U.; Farooq, U.; Halim, S.A.; Khan, A.; Al-Harrasi, A. Recent advances in combinatorial cancer therapy via multifunctionalized gold nanoparticles. Nanomedicine 202015, 1221–1237.

-Corti, A.; Sacchi, A.; Gasparri, A.M.; Monieri, M.; Anderluzzi, G.; Colombo, B.; Gori, A.; Mondino, A.; Curnis, F. Enhancement of doxorubicin anti-cancer activity by vascular targeting using IsoDGR/cytokine-coated nanogold. J. Nanobiotechnol. 202119, 128.

-The authors should add table in the sections 2 that summarize the specific role of the type of metal, core size, stability, aggregation state, global charge given by zeta potential and type of functionalization in the specific cancer treatment. Moreover, a comparative final discussion is needed on the basis of the result given in this table.

-The conclusion section should be improved summarizing the result given in the proposed table and comparing the different type of nanosystems discussed in this review.

I consider that the quality of the English is appropiate and only needs minor editing.

Author Response

Based on your comments and the observations of the other referee, we have modified the manuscript.

Point 1: In the introduction section the reason for selecting palladium, gold and silver nanoparticles among others metallic nanoparticles is needed to be included.

Response 1: The manuscript has been changed, lines 70-73:

Point 2: Page 2, line 48-49. The definition of nanoparticles is incorrect. In fact, NPs can have a size that ranges from 1 nm to 100 nm. See for instance:

Response 2: : Manuscript has been modified to include recommended bibliography line 55

Point 3: In section 2, metallic nanoparticles, the authors have forgotten to add comment or mention to the important role of the size of nanoparticles in the cell internalization, toxicity and the protection against enzymatic degradation exerted to biomolecules like DNA or RNA. See for instance:

Thorek, D.L.; Tsourkas, A. Size, charge and concentration dependent uptake of iron oxide particles by non-phagocytic cells. Biomaterials 200829, 3583–3590

Ghosh, P.S.; Kim, C.K.; Han, G.; Forbes, N.S.; Rotello, V.M. Efficient gene delivery vectors by tuning the surface charge density of amino acid-functionalized gold nanoparticles. ACS Nano 20082, 2213–2218

Response 3: : Manuscript has been modified to include recommended bibliography, Lines 84-95

Point 4: In section 2.2 about the role of functionalized gold nanoparticles in cancer therapy, the authors have forgotten some recent important works. A comment about minimization of side effects of functionalized gold nanoparticles and different nanoformulation is needed. See for instance:

Giráldez-Pérez, R.M.; Grueso, E.; Montero-Hidalgo, A.J.; Luque, R.M.; Carnerero, J.M.; Kuliszewska, E.; Prado-Gotor, R. Gold Nanosystems Covered with Doxorubicin/DNA Complexes: A Therapeutic Target for Prostate and Liver Cancer. Int. J. Mol. Sci. 202223, 15575.

Giráldez-Pérez, R.M.; Grueso, E.; Domínguez, I.; Pastor, N.; Kuliszewska, E.; Prado-Gotor, R.; Requena-Domenech, F. Biocompatible DNA/5-Fluorouracil-Gemini Surfactant-Functionalized Gold Nanoparticles as Promising Vectors in Lung Cancer Therapy. Pharmaceutics 202113, 423.

Ahmad, T.; Sarwar, R.; Iqbal, A.; Bashir, U.; Farooq, U.; Halim, S.A.; Khan, A.; Al-Harrasi, A. Recent advances in combinatorial cancer therapy via multifunctionalized gold nanoparticles. Nanomedicine 202015, 1221–1237.

Corti, A.; Sacchi, A.; Gasparri, A.M.; Monieri, M.; Anderluzzi, G.; Colombo, B.; Gori, A.; Mondino, A.; Curnis, F. Enhancement of doxorubicin anti-cancer activity by vascular targeting using IsoDGR/cytokine-coated nanogold. J. Nanobiotechnol. 202119, 128.

Response 4: Manuscript has been modified to include recommended bibliography, Line 255-293

Point 5: The authors should add table in the sections 2 that summarize the specific role of the type of metal, core size, stability, aggregation state, global charge given by zeta potential and type of functionalization in the specific cancer treatment. Moreover, a comparative final discussion is needed on the basis of the result given in this table.

Response 4: The manuscript was changed based on the comments so that Table 1 was included.

Dear referee, we greatly appreciate all your comments, we hope that our answers are adequate, so that you consider our article for publication. Best regards.

Reviewer 2 Report

Referee Report

Title: Functionalized Metal Nanoparticles in Cancer Therapy

Manuscript ID: pharmaceutics-2470884

By Villalobos-Gutiérrez et al

Submitted to Pharmaceutics (ISSN 1999-4923)

  1. Abstract: The authors provided background information and the rationale for conducting the review study. However, the abstract lacks the mention of results, future prospects, and a conclusion.
  2. Lines 31-33: The statement "antioxidants, radionics, and nanoparticles" does not accurately represent cancer therapy modalities, as they are materials/agents used in cancer therapy. Please rephrase this statement accordingly.
  3. Lines 43-46: When discussing the use of gold nanomaterials in cancer therapy, please include more recent references such as Siddique et al. (Applied Sciences 2020;10:3824) and Moore et al. (Nano Express 2021;2:022001).
  4. Lines 55-58: When referring to metallic NPs for biomedical applications, please consider incorporating more updated references such as Siddique et al. (Nanomaterials 2022;12:2826).
  5. It would be beneficial to include a table summarizing the various applications of NPs in cancer therapy.
  6. The addition of NPs can also enhance important therapy modalities such as radiotherapy, for example, gold nanoparticle-enhanced radiotherapy. The authors should mention relevant studies on this topic, such as Chow et al. (Micromachines 2023;14:1230) and Sadiq et al. (Nanomaterials 2022;12:2991).
  7. It is recommended to include a section on future prospects, highlighting potential directions for future research in this field.

Author Response

Based on your comments and the observations of the other referee, we have modified the manuscript.

Point 1: Abstract: The authors provided background information and the rationale for conducting the review study. However, the abstract lacks the mention of results, future prospects, and a conclusion.

Response 1: The manuscript was modified, line: 19-25

Point 2: Lines 31-33: The statement "antioxidants, radionics, and nanoparticles" does not accurately represent cancer therapy modalities, as they are materials/agents used in cancer therapy. Please rephrase this statement accordingly.

Response 2: The manuscript was modified, line 38-41

Point 3: Lines 43-46: When discussing the use of gold nanomaterials in cancer therapy, please include more recent references such as Siddique et al. (Applied Sciences 2020;10:3824) and Moore et al. (Nano Express 2021;2:022001).

Response 3: The references in these lines are 8, 9 and 10, and they are recent:

Gan, W.W.; Chan, L.W.; Li,W.; Wong, T.W. Critical clinical gaps in cancer precision nanomedicine development. J. Control. Release. 2022, 345, 811–818. https://doi.org/10.1016/j.jconrel.2022.03.055

Mundekkad, D.; Cho, W.C. Nanoparticles in Clinical Translation for Cancer Therapy. Int. J. Mol. Sci. 2022, 23, 1685. https://doi.org/10.3390/ijms23031685

Cheng, Z.; Li, M.; Dey, R.; Chen, Y. Nanomaterials for cancer therapy: current progress and perspectives. J Hematol Onco.l 2021, 14, 85. https://doi.org/10.1186/s13045-021-01096-0

However, we consider the inclusion of Moore 2021 in lines 161-162; 168-171.

Point 4: Lines 55-58: When referring to metallic NPs for biomedical applications, please consider incorporating more updated references such as Siddique et al. (Nanomaterials 2022;12:2826).

Response 4: The manuscript was modified, line 64-67.

Point 5: It would be beneficial to include a table summarizing the various applications of NPs in cancer therapy.

Response 5: The table was created (Table 1)

Point 6: The addition of NPs can also enhance important therapy modalities such as radiotherapy, for example, gold nanoparticle-enhanced radiotherapy. The authors should mention relevant studies on this topic, such as Chow et al. (Micromachines 2023;14:1230) and Sadiq et al. (Nanomaterials 2022;12:2991).

Response 6: Based on the inclusion and exclusion criteria of the bibliographic review, which includes in vitro and in vivo studies, we have decided that the articles you recommended do not fall within the inclusion criteria, but we still appreciate your comments.

Point 7: It is recommended to include a section on future prospects, highlighting potential directions for future research in this field.

Response 7: Future perspectives has been included: Lines 485-499

Dear referee, we greatly appreciate all your comments, we hope that our answers are adequate, so that you consider our article for publication. Best regards.

Reviewer 3 Report

Please find attached the report file

Please find attached the report file

Author Response

Based on your comments and the observations of the other referee, we have modified the manuscript.

The manuscript titled “Functionalized metal nanoparticles in cancer therapy” provides a comprehensive overview of the current landscape of cancer treatment, highlighting the limitations of conventional therapies and the potential of nanotechnology-based approaches. The authors discuss the rising burden of cancer worldwide and the need for more effective and targeted treatment strategies. The manuscript then delves into the various types of nanoparticles used in cancer therapy, emphasizing their advantages and physicochemical properties that influence their biological effects. Overall, the manuscript is well-structured and provides a valuable contribution to the field of cancer research. However, there are a few areas that could be improved or clarified.

Point 1: The overall language and clarity of the manuscript are good. However, some sentences could be rephrased to improve readability and avoid repetition. It is advised to carefully proofread the manuscript to correct any grammatical errors, typographical mistakes, or awkward sentence constructions..

Response 1: The manuscript was reviewed again, making the necessary corrections.

Point 2: The discussion on the physicochemical properties of nanoparticles, including shape, size, surface charge, and degree of dispersion, is crucial in understanding their biological effects. However, it would be beneficial to elaborate on the impact of these properties on specific aspects of cancer therapy, such as cellular uptake, targeting efficiency, and biodistribution.

Response 2: The manuscript was modified, line 87-95

Point 3:      The “2.2 Gold Nanoparticles” section should provide a broad overview of the different PTT applications for tumor eradication. I would suggest further expanding the discussion of the various cutting-edge study by including some others examples of gold nanoparticles capped with protein like : ( Biomimetic keratin gold nanoparticle-mediated in vitro photothermal therapy on glioblastoma multiforme - https://doi.org/10.2217/nnm-2020-0349 ).

Response 3: The manuscript was modified, the recommended bibliography was inserted in line 294-302. And a new bibliography has been included (https://doi.org/10.1039/d0tb00240b), lines 302-313.

Point 4: The manuscript would benefit from the inclusion of recent advancements and promising developments in the field of nanoparticle-based cancer therapy. This could include emerging nanotechnologies, such as gene delivery systems, immunotherapy, or combination therapies, to provide a more comprehensive and up-to-date review. Overall, the manuscript provides a valuable review of the use of nanoparticles in cancer therapy. The incorporation of recent advancements and a more detailed exploration of the impact of physicochemical properties would further enhance its quality. The manuscript is well-structured, and the information presented is relevant and supported by appropriate references. With minor revisions, this review has the potential to be an informative and valuable resource in the field of cancer research

Response 4: Studies on delivery systems, immunotherapy, chemotherapy, and photothermal therapy were included in the manuscript and are summarized in Table 1.

Dear referee, we greatly appreciate all your comments, we hope that our answers are adequate, so that you consider our article for publication. Best regards.

Reviewer 4 Report

The presented manuscript, which is a review of data/results of studies on the possibility of using metal nanoparticles in the diagnosis/treatment of cancer, lacks a critical view of this problem. Authors, in addition to collecting literature data, should evaluate the pros and cons of such procedures and indicate the most interesting/promising (from the point of view of clinical practice) directions for further research. A review article should be an original look at the problem, not just a collection of data.

Author Response

Based on your comments and the observations of the other referee, we have modified the manuscript.

Point 1: The presented manuscript, which is a review of data/results of studies on the possibility of using metal nanoparticles in the diagnosis/treatment of cancer, lacks a critical view of this problem. Authors, in addition to collecting literature data, should evaluate the pros and cons of such procedures and indicate the most interesting/promising (from the point of view of clinical practice) directions for further research. A review article should be an original look at the problem, not just a collection of data.

Response 1: Thank you for the comments. The Future Perspective section was included in the manuscript addressing the suggested comments, line: 19-25; 495-511. A table has been included summarizing all the studies consulted.

Dear referee, we greatly appreciate all your comments, we hope that our answers are adequate, so that you consider our article for publication. Best regards.

Round 2

Reviewer 1 Report

The authors have revised and improved the manuscript in accordance with the suggestions made. The paper is now worth of being published in Pharmaceutics.

Reviewer 2 Report

The authors have addressed all my concerns in this revision.